# Dengue in New Caledonia: Knowledge and Gaps

**DOI:** 10.3390/tropicalmed4020095

**Published:** 2019-06-20

**Authors:** Catherine Inizan, Arnaud Tarantola, Olivia O’Connor, Morgan Mangeas, Nicolas Pocquet, Carole Forfait, Elodie Descloux, Ann-Claire Gourinat, Anne Pfannstiel, Elise Klement-Frutos, Christophe Menkes, Myrielle Dupont-Rouzeyrol

**Affiliations:** 1Institut Pasteur de Nouvelle-Calédonie, Institut Pasteur International Network, URE Dengue et Arboviroses, Noumea 98845, New Caledonia; cinizan@pasteur.nc (C.I.); ooconnor@pasteur.nc (O.O.); 2Institut Pasteur de Nouvelle-Calédonie, Institut Pasteur International Network, Unité d’Epidémiologie, Noumea 98845, New Caledonia; atarantola@pasteur.nc; 3Institut de Recherche pour le Développement, UMR ENTROPIE 9220. IRD, CNRS, UR, Noumea 98800, New Caledonia; morgan.mangeas@ird.fr (M.M.); christophe.menkes@ird.fr (C.M.); 4Institut Pasteur de Nouvelle-Calédonie, Institut Pasteur International Network, URE Entomologie médicale, Noumea 98845, New Caledonia; npocquet@pasteur.nc; 5Direction des Affaires Sanitaires et Sociales de Nouvelle-Calédonie, Service de Santé Publique, Noumea 98800, New Caledonia; carole.forfait@gouv.nc (C.F.); anne.pfannstiel@gouv.nc (A.P.); 6Centre Hospitalier Territorial Gaston-Bourret, Service de Médecine interne, Dumbea-Sur-Mer 98835, New Caledonia; desclouxe@gmail.com (E.D.); elise.klement@cht.nc (E.K.-F.); 7Centre Hospitalier Territorial Gaston-Bourret, Laboratoire de Microbiologie, Dumbea-Sur-Mer 98835, New Caledonia; ann-claire.gourinat@cht.nc

**Keywords:** arbovirus, epidemiology, risk assessment, Pacific, climate

## Abstract

Arboviruses are viruses transmitted to humans by the bite of infected mosquito vectors. Over the last decade, arbovirus circulation has increasingly been detected in New Caledonia (NC), a French island territory located in the subtropical Pacific region. Reliable epidemiological, entomological, virological and climate data have been collected in NC over the last decade. Here, we describe these data and how they inform arboviruses’ epidemiological profile. We pinpoint areas which remain to be investigated to fully understand the peculiar epidemiological profile of arbovirus circulation in NC. Further, we discuss the advantages of conducting studies on arboviruses dynamics in NC. Overall, we show that conclusions drawn from observations conducted in NC may inform epidemiological risk assessments elsewhere and may be vital to guide surveillance and response, both in New Caledonia and beyond.

## 1. New Caledonia as A Near-isolated Observation Site

In the past decade, arbovirus co-circulation has increasingly been detected in New Caledonia (NC), a French island territory located in the subtropical Pacific region with a population of approximately 280,000 (Figure 1A). In countries where arboviruses are endemic, the complexity of ecosystems impedes the identification of the determinants of arbovirus circulation and their relative contribution. The insular nature of NC, the presence of only one known arbovirus vector (*Aedes aegypti*), the relatively low co-circulation of several arboviruses, the marked seasonality and the reliable epidemiological and entomological surveillance systems reduce this complexity, establishing NC as a simple and relevant ecosystem where these determinants and their interactions can be thoroughly analyzed. Here, we present the epidemiological, entomological, virological and climate data collected in NC over the last decade and how they inform arboviruses’ epidemiological profile. We also discuss the advantages of conducting studies on arboviruses dynamics in NC.

## 2. Dengue Circulation was Detected Every Year in New Caledonia over the Last Decade

Epidemiological data indicate that dengue virus (DENV) is the major arbovirus circulating in the Pacific region [1]. Over earlier decades, DENV circulation in NC displayed a 3–4 year cyclical pattern of occurrence, with two epidemic peaks occurring during two consecutive hot and rainy seasons. Few cases were detected, whether between outbreaks or during the cool season between epidemic peaks during epidemic years [2]. Since 2008 however, DENV circulation has increased in NC, causing recurrent outbreaks with cases detected every year. DENV caused three major outbreaks in NC in 2008–2009, 2012–2013 and 2016–2018, leading to 9589, 11,240, and 7266 reported cases, respectively (https://dass.gouv.nc/votre-sante/documents-rapports-etudes). DENV-1 circulated uninterruptedly during those periods, co-circulating with DENV-4 in 2009, DENV-3 in 2014, DENV-2 and -3 in 2017 and DENV-2 in 2018 (Figure 1B) (https://dass.gouv.nc/votre-sante/documents-rapports-etudes). Molecular surveillance showed that DENV-1-genotype I replaced DENV-1-genotype IV in 2012 [2]; this genotype replacement was followed by a large outbreak in 2012–2013. Further, the 2016–2018 dengue outbreak caused more clinically severe cases compared to previous years. Since 2012, we have observed major changes in the epidemiology of dengue, with uninterrupted DENV circulation causing cases detected every year and even during the cool season. Along with DENV circulation, Zika virus (ZIKV) was detected in NC in 2014, causing a significant outbreak of 1392 confirmed cases [3] (https://dass.gouv.nc/votre-sante/documents-rapports-etudes) (Figure 1B). Despite repeated introductions, chikungunya virus (CHIKV) has not caused major outbreaks in NC, with less than 50 cases detected in 2011, 2013, 2014 and 2015 [1] (https://dass.gouv.nc/votre-sante/documents-rapports-etudes) (Figure 1B). No signs of circulation of CHIKV or ZIKV were detected prior to these outbreaks.

## 3. Impact of Population Immunity on the Arbovirus Circulation Profile in New Caledonia

In NC, the percentage of the population protected against arboviruses is unknown. Population immunity shapes the epidemiological profile of arbovirus circulation. Introduction of an arbovirus in an immunologically naïve population favours outbreak occurrence. Unlike that of ZIKV, however, the introduction of CHIKV did not lead to major outbreaks. DENV-1 has been circulating continuously since 2008; as the low net migration (22,500 new inhabitants between 2009 and 2014, http://www.isee.nc/population/demographie/migrations) along with moderate fertility (15.4‰ in 2016, http://www.isee.nc/population/demographie/naissances-fecondite) are unlikely to lead to dramatic changes in population immunity, a protective population immunity threshold has probably not been reached, as outbreaks occurred in 2013–2014 and in 2016–2018.

## 4. Contribution of Entomological Parameters to Arbovirus Epidemiology in New Caledonia

Effective entomological surveillance (http://www.institutpasteur.nc/wp-content/uploads/2016/12/) identified *Ae. aegypti* as the main vector for arboviruses in NC. The Larvae and Pupae Index (LPI) is a proxy for *Ae. aegypti* density and was calculated as the average number of late-instar larvae and pupae per house for 300 randomly selected houses visited per month. LPI remained low from 2010 to 2016 in Noumea (http://www.institutpasteur.nc/wp-content/uploads/2016/12/) (Figure 1C) but a dengue outbreak occurred in 2013 nevertheless. As documented elsewhere [4], vector densities therefore seem to play a limited role in conditioning the occurrence of arbovirus outbreaks in NC. 

Could outbreaks be determined by the ability of available vectors to transmit the virus? Vector competence studies showed that *Ae. aegypti* from NC is competent for DENV, ZIKV and CHIKV. The Extrinsic Incubation Period, however, is three days for CHIKV [5], seven days for DENV [6] and nine days for ZIKV [7]. Furthermore, transmission efficiencies at 14 days post-infection are 30% for CHIKV [5], <10% for DENV [6] and 5% for ZIKV [7]. Although transmission efficiency was highest for CHIKV, this virus never caused major outbreaks in NC.

## 5. Cimate Oscillation is An Important Parameter to Forecast Dengue Outbreaks in New Caledonia

The climate of NC can be described as tropical (Figure 1A) with a hot and rainy season (November to April) with mean monthly temperatures varying between 25–28 °C and mean monthly precipitation varying spatially between 3 to 6 mm/day, followed by a cooler and drier season (May to October) with monthly temperature between 22 and 25 °C and precipitation between 1 to 3 mm/day. Figure 1A shows precipitations and air temperatures. Precipitation data were generated from the CPC Merged Analysis of Precipitation (CMAP) data available at https://climatedataguide.ucar.edu/climate-data/cmap-cpc-merged-analysis-precipitation. Air temperature were taken from https://www.esrl.noaa.gov/psd/data/gridded/data.ncep.reanalysis2.html. The monthly data initially covering 1979–2018 were used to calculate a monthly seasonal cycle of surface precipitation and temperature and Figure 1 A represents the two seasons, summer (November to April, left) and winter (May to October, right) calculated from those climatology data. At the local scale, NC is a mountainous island also presenting specially contrasted temperature and precipitation between the central mountainous (coolest) areas and its coasts. The west coast is dryer and warmer than the east coast [8,9]. In that region, the El Niño-Southern Oscillation (ENSO) is a major cause of climate interannual variations [10], with NC experiencing warmer (~+1 °C) and wetter (~+20%) summers during La Niña years. While the number of Western Pacific Islands with dengue epidemics has been shown to increase during La Niña years [11], there is no established link between ENSO and the occurrence of dengue epidemics in NC [12].

In a context of global climate change, seasonality in NC is expected to change in the region [13], resulting in longer and warmer but dryer summer seasons. Seasonal variations of climate at the country scale may therefore represent an important determinant of arbovirus outbreaks with adverse effects on viruses and vectors as both temperature and precipitations have been shown to be strongly influence the occurrence of dengue epidemics in NC [9,12].

## 6. How Can We Better Understand and Forecast Arbovirus Outbreaks?

The time of year at which the first arbovirus cases are detected on the territory is likely important. The risk of outbreak might be greater when an imported arbovirus is introduced during a favorable time of year.

Interestingly, DENV and ZIKV are both flaviviruses and both caused outbreaks in NC. Neither CHIKV over the last decade nor Ross River Virus (RRV) in the late seventies [14]—both alphaviruses—caused outbreaks in NC. Flaviviruses and alphaviruses may thus behave very differently in the NC context. 

Genotype, clade and strain replacements may impact DENV epidemiological profile by modulating viral replicative fitness [15]. As DENV serotypes are not antigenically homogeneous [16], it can also be hypothesized that infection with one genotype does not confer complete cross-protection against other genotypes. DENV-1 genotype replacement in 2012 [2] may thus have contributed to DENV-1 serotype’s uninterrupted circulation over the last decade in NC.

A better and more complete understanding of the peculiar epidemiological profile of arbovirus circulation in NC and its determinants can improve risk evaluation and public health planning before sizeable outbreaks. Although fragmented and incomplete, data collected over the last decade enabled a better description and understanding of this epidemiology. From these observations, we can conclude that predicting the occurrence of arbovirus outbreaks in NC relies on multiple factors. Measuring their interactions using mathematical models is of prime importance to identify new combinations of parameters to take into account when attempting to decipher the causes leading to an arbovirus outbreak. Modelling may help identify and understand the mechanisms that underlie the dynamics and evolution of arboviruses epidemiological profile in NC. Further, a deeper understanding of vector biology and population dynamics along with clinical and in vitro virological studies will generate highly informative data, to inform and strengthen these models. Finally, collecting more reliable data with higher spatial resolution on population immunity, vector densities, temperature/humidity oscillations, geolocalisation and time of cases’ introduction, genetic evolution and replicative fitness of the arbovirus introduced in its human and mosquito hosts will provide valuable input to build robust predictive models.

## 7. Conclusions

New Caledonia is an island with limited and documented population exchanges and no sustained introduction of arboviruses cases. Its modest population size, effective surveillance, health and research systems, and the presence of only one known arbovirus vector make NC an ideal observatory of factors modulating the epidemiology of arboviruses and their differences. 

Conclusions drawn from these observations may inform epidemiological risk assessments elsewhere—especially in non-endemic areas where the vector is present—and may be vital to guide surveillance and responses, both in New Caledonia and beyond. The World Mosquito Program will be implemented in Noumea in July 2019. The release of *Wolbachia*-treated *Ae. aegypti* will reshape the epidemiology of arbovirus circulation in NC. Understanding arbovirus epidemiology and its determinants before this implementation will provide a highly valuable baseline to document the effects of the *Wolbachia* strategy. 

## Figures and Tables

**Figure 1 tropicalmed-04-00095-f001:**
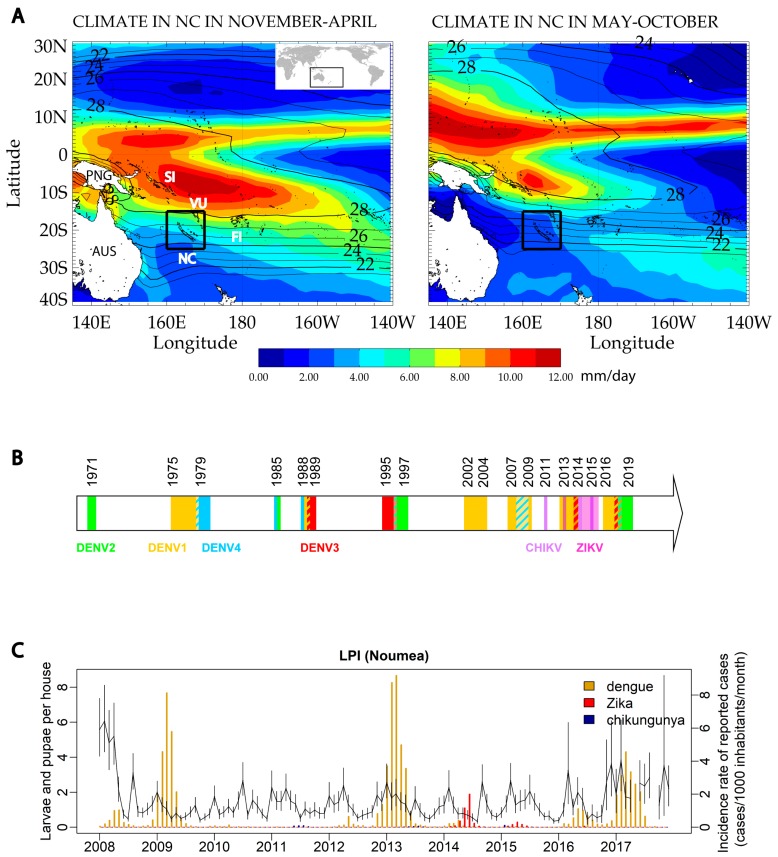
Temperatures, precipitations and history of arbovirus circulation and vector densities in New Caledonia, South Pacific region. (**A**) Summertime and wintertime maps of precipitation (rainbow scale, mm/day) with contours of air temperature (°C). PNG: Papua-New Guinea, AUS: Australia, SI: Solomon Islands; FI: Fiji, NC: New Caledonia. (**B**) Timeline of arboviruses circulation in New Caledonia between 1971 and 2018 (Source Institut Pasteur in New Caledonia, generated based on the data published by the Health Authorities (https://dass.gouv.nc/votre-sante/documents-rapports-etudes)). (**C**) Graphical representation of the monthly average numbers of larvae and pupae per house in Noumea between 2008 and 2018. Error bars indicate the 95% confidence intervals. Incidence rates expressed as the number of cases per 1000 inhabitants per month are indicated for dengue, Zika and chikungunya (http://www.institutpasteur.nc/wp-content/uploads/2016/12/).

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
