# Peer review of "Dengue in New Caledonia: Knowledge and Gaps"

_tropicalmed, 2019, doi:10.3390/tropicalmed4020095_

Round 1
Reviewer 1 Report
This is a well-written short review on epidemiology of mosquito-borne viral diseases in New Caledonia island. The authors provide interesting and up-to-dated epidemiological data on circulation of dengue, chikungunya, and zika viruses in the French island territory of subtropical Pacific region. The impact of climate change on the risk of arbovirosis in New Caledonia is raised. The authors mention that implementation of Wolbachia strategy for prevention of arboviruses would necessitate an improved knowledge of virus dynamics in New Caledonia. The authors believe that this French island territory could be informative on epidemiology of arboviruses circulation in tropical island regions.
Author Response
We thank the reviewer for carefully reviewing our article and the kind comments.

Reviewer 2 Report
The manuscript “Dengue in New-Caledonia: knowledge and gaps” by Inizan et al present a very interesting perspective on dengue epidemiology, along with entomological and virologic data.
Some kind of functional sub-headings might help the reader (though I am not sure if this format is specified by the journal). For example, the first block is “New Caledonia as a near isolated observation site”, followed by “Dengue epidemic cycle is persistent over decades”…
Fig 1 b – Its surprising to see that no DENV-2 was observed in NC between 2016-2018, while that was the predominant type in nearby most populated tropical countries like Srilanka, India, china, and likely Thailand, Malayasia and Indonesia. Also, interesting that Zika managed to cause an epidemic there.
Fig 1C – please assign a different, more intense color for the chikungunya data. They are not distinguishable currently.
Please provide additional information regarding how the data were collected. It does not seem like all data presented here came from the references (example : 1392 confirmed zika cases). When you are not referring to a source please provide some information regarding the methods, assuming they are your data. Additionally, please place the data source in figure legends.
Do you have any information regarding dengue re-infection ? Any instance of re-infection by same serotype ??
Author Response
The manuscript “Dengue in New-Caledonia: knowledge and gaps” by Inizan et al present a very interesting perspective on dengue epidemiology, along with entomological and virologic data.
We thank the reviewer for the insightful comments. We answer to the points raised by the reviewer below.
Comment 1:
Some kind of functional sub-headings might help the reader (though I am not sure if this format is specified by the journal). For example, the first block is “New Caledonia as a near isolated observation site”, followed by “Dengue epidemic cycle is persistent over decades”…
Authors’ reply to Comment 1:
We thank the reviewer for this suggestion. Sub-headings were added to the text, which indeed provides clarity and structure to the article.
Comment 2:
Fig 1 b – Its surprising to see that no DENV-2 was observed in NC between 2016-2018, while that was the predominant type in nearby most populated tropical countries like Sri Lanka, India, China, and likely Thailand, Malaysia and Indonesia. Also, interesting that Zika managed to cause an epidemic there.
Authors’ reply to Comment 2:
The reviewer is correct in underlining that DENV-2 was the major circulating serotype in the region between 2016 and 2018. DENV-2 was also detected in NC over that period. We indicate in the text that DENV-1 co-circulated with “DENV-2 and -3 in 2017 and DENV-2 in 2018”.
DENV-1 circulation was uninterrupted in NC over the last decade, while it co-circulated with other serotypes.
Comment 3:
Fig 1C – please assign a different, more intense color for the chikungunya data. They are not distinguishable currently.
Authors’ reply to Comment 3:
Chikungunya virus caused very few cases in NC, with less than 50 cases detected in 2011, 2013, 2014 and 2015, leading to an incidence rate of approximately 0.18/1,000 inhabitants. The data are therefore not distinguishable in the graph.
As per the reviewer’s request we nevertheless changed the color of chikungunya incidence rate of reported cases to make it more clear that chikungunya circulation was almost below the detection limit in NC over the last decade.
Comment 4:
Please provide additional information regarding how the data were collected. It does not seem like all data presented here came from the references (example : 1392 confirmed zika cases). When you are not referring to a source please provide some information regarding the methods, assuming they are your data. Additionally, please place the data source in figure legends.
Authors’ reply to Comment 4:
The two sources are different.
Data related to the number of arbovirosis cases originate from epidemiological surveillance conducted over the last decade under the supervision of the local Health Authorities. This source is quoted in the text.
Entomological indices arise from entomological surveillance networks conducted by the Institut Pasteur in New Caledonia. Larvae and Pupae Indices (LPI) for Noumea between 2008 and 2017 were published on the Institut Pasteur in New Caledonia website (http://www.institutpasteur.nc/wp-content/uploads/2016/12/) and this source is quoted in the text.
Maps of rainfall in NC were generated for Figure 1A and have never been published before.
Actions taken:
- Hyperlinks of source data have been added to the text and figure legends
- We have added a sentence in the text indicating how LPI was calculated :
“The Larvae and Pupae Index (LPI) is a proxy for Ae. aegypti density and was calculated as the average number of late-instar larvae and pupae per house for 300 randomly selected houses visited per month.”
- We have added a sentence in the text describing how maps of precipitation and air temperature were generated
“Figure 1A shows precipitations and air temperatures. Precipitation were generated from the CMAP data available at https://climatedataguide.ucar.edu/climate-data/cmap-cpc-merged-analysis-precipitation. Air temperature were taken from https://www.esrl.noaa.gov/psd/data/gridded/data.ncep.reanalysis2.html. The monthly data initially covering 1979-2018 were used to calculate a monthly seasonal cycle of surface precipitation and temperature and Figure 1 A represents the two seasons, summer (NDJFMA, left) and winter (MJJASMO, right) calculated from those climatology. “
Comment 5:
Do you have any information regarding dengue re-infection ? Any instance of re-infection by same serotype ?
Authors’ reply to Comment 5:
Although re-infection by the same dengue serotype has been documented elsewhere (Waggoner, J Infect Dis, 2016; Forshey, PLoS Negl Trop Dis, 2016), homotypic re-infections have never been detected in New Caledonia. This could arise from the challenges in linking patients across epidemic years. Further, dengue became a notifiable disease in New Caledonia in 1991. Cases prior to this date were probably under-reported, thus impairing the detection of re-infection. Finally, and most importantly, as dengue infection can be inapparent or lead to mild symptoms, it is likely that patients did not seek medical care for either primary or secondary dengue infection and re-infections were thus not detected.
